# Single-Dose Treatment with Rapamycin Preserves Post-Ischemic Cardiac Function through Attenuation of Fibrosis and Inflammation in Diabetic Rabbit

**DOI:** 10.3390/ijms24108998

**Published:** 2023-05-19

**Authors:** Arun Samidurai, Manu Saravanan, Ramzi Ockaili, Donatas Kraskauskas, Suet Ying Valerie Lau, Varun Kodali, Shakthi Ramasamy, Karthikeya Bhoopathi, Megha Nair, Sean K. Roh, Rakesh C. Kukreja, Anindita Das

**Affiliations:** Division of Cardiology, Pauley Heart Center, Internal Medicine, Virginia Commonwealth University, Richmond, VA 23298, USA; arun.samidurai@vcuhealth.org (A.S.);

**Keywords:** diabetes, mTOR, rapamycin, ischemia/reperfusion injury, fibrosis, inflammation

## Abstract

Robust activation of mTOR (mammalian target of rapamycin) signaling in diabetes exacerbates myocardial injury following lethal ischemia due to accelerated cardiomyocyte death with cardiac remodeling and inflammatory responses. We examined the effect of rapamycin (RAPA, mTOR inhibitor) on cardiac remodeling and inflammation following myocardial ischemia/reperfusion (I/R) injury in diabetic rabbits. Diabetic rabbits (DM) were subjected to 45 min of ischemia and 10 days of reperfusion by inflating/deflating a previously implanted hydraulic balloon occluder. RAPA (0.25 mg/kg, i.v.) or DMSO (vehicle) was infused 5 min before the onset of reperfusion. Post-I/R left ventricular (LV) function was assessed by echocardiography and fibrosis was evaluated by picrosirius red staining. Treatment with RAPA preserved LV ejection fraction and reduced fibrosis. Immunoblot and real-time PCR revealed that RAPA treatment inhibited several fibrosis markers (TGF-β, Galectin-3, MYH, p-SMAD). Furthermore, immunofluorescence staining revealed the attenuation of post-I/R NLRP3-inflammasome formation with RAPA treatment as shown by reduced aggregation of apoptosis speck-like protein with a caspase recruitment domain and active-form of caspase-1 in cardiomyocytes. In conclusion, our study suggests that acute reperfusion therapy with RAPA may be a viable strategy to preserve cardiac function with the alleviation of adverse post-infarct myocardial remodeling and inflammation in diabetic patients.

## 1. Introduction

Despite the considerable advancement in therapeutic strategies, diabetes still continues to present a risk of adverse cardiovascular events after myocardial ischemia/reperfusion (I/R) injury, ultimately causing heart failure and even cardiogenic death [1]. Diabetes itself plays an important role in the pathogenesis of myocardial fibrosis, predominantly causing ventricular remodeling and ultimately inducing heart failure [2,3,4]. The inhibition of the mammalian target of rapamycin (mTOR) pathway with rapamycin (RAPA, mTOR inhibitor) has shown promising results in controlling diabetes-induced detrimental effects during acute myocardial I/R injury [5,6,7]. mTOR, a serine-threonine kinase, is a master regulator of cellular metabolism and growth which also monitors nutrient availability and energy utilization. mTOR serves as the catalytic core of two unique complexes, mTORC1 and mTORC2, which are differentiated by the integration of the key accessory complexes Raptor and Rictor, respectively. The shared mTOR catalytic core subunit is composed of the DEPTOR (DEP domain containing mTOR-interacting protein), the Tti1-Tel2 complex, and the mLST8 (mammalian lethal with sec-13 protein 8) [8]. mTORC1 is strongly inhibited by RAPA, while its counterpart mTORC2 is more resistant due to its RAPA-insensitive companion of mTOR, Rictor.

Our previous work has shown that RAPA’s inhibitory effect on mTORC1 significantly reduced infarct size, necrosis, and apoptosis of cardiomyocytes after myocardial I/R injury [9,10,11]. In addition, the cardioprotective effects of RAPA is associated with the signaling pathways including phosphorylation of STAT3, ERK and endothelial NOS. These factors assist in inactivating GSK-3β while bolstering the anti-apoptotic Bcl-2 to Bax ratio [9,10]. mTORC2 can also be selectively activated with simultaneous inhibition of mTORC1 by RAPA in limiting MI (myocardial infarction)-induced cardiomyocyte necrosis and apoptosis [12]. We showed that chronic administration of RAPA preferentially inhibited mTORC1, alleviated oxidative stress, and reduced cardiac dysfunction with alteration of the expression of antioxidant and contractile proteins in type 2 diabetic mice [13]. Likewise, specific mTORC1 inhibition by overexpression of PRAS40 or PF-4708671 treatment improved metabolic and cardiac function in diabetic or diet-induced obese mice [14,15]. Moreover, treatment with RAPA at the onset of reperfusion reduces myocardial infarct size following 45 min of ischemia and 3 days of reperfusion in diabetic rabbits by inhibiting mTORC1, while restoring mTORC2 [16].

Poor prognosis as well as higher mortality of diabetic patients after MI are associated with the frequent development of subsequent pathological myocardial remodeling and inflammation with concomitant cardiac dysfunction [17]. It has been shown that transforming growth factor β (TGF-β) signaling, one of the powerful markers of fibrosis, directly induces mTORC1 in cardiac fibroblasts, invoking increased collagen production. mTORC1 propagates fibrogenesis through the activation of ATF4 (activating transcription factor 4) and PFKB3 (6-phosphofructo-2-kinase/fructose-2,6-bisphosphatase 3), thereby metabolically driving myofibroblasts to supply substrates for collagen synthesis [18]. Importantly, inhibition of mTORC1 with RAPA significantly attenuated collagen synthesis and deposition in the myocardium of transgenic mice overexpressing heparin-binding epidermal growth factor (HN-EGF) [19]. Furthermore, the role of mTOR in cardiac inflammation is also critical, as its inhibition has been shown to decrease inflammation and reduce proteasomal degradation with attenuation of infarction and adverse cardiac remodeling after MI in rats [20].

Based on this background information, we contemplated that treatment with RAPA at reperfusion could preserve cardiac function through the attenuation of cardiac fibrosis and cryopyrin (nucleotide-binding oligomerization domain-like receptor pyrin domain containing-3, NLRP3)-mediated inflammation in diabetic rabbits subjected to myocardial I/R injury in conscious state. Our results provide compelling evidence that mTOR inhibition with a single dose of RAPA at reperfusion has long-lasting effect in protecting the diabetic myocardium against I/R-associated fibrotic and inflammatory mediators of injury.

## 2. Results

### 2.1. Rapamycin Improves Post-Ischemic Cardiac Function

Figure 1 shows the experimental protocol in the rabbit model of conscious I/R injury. We measured cardiac function, fibrosis, cell size and expression of different markers of fibrosis and inflammasome formation. The results showed that post-ischemic cardiac function was significantly improved in the RAPA-treated diabetic rabbits (DM-I/R + RAPA) as compared to vehicle-treated controls (DM-I/R) (Figure 2). The left ventricular ejection fraction (LVEF: 64 ± 1.4%), left ventricular fractional shortening (LVFS: 37 ± 0.9%), stroke volume (SV: 903 ± 100 µL) and cardiac output (CO: 284 ± 38 mL/min) were significantly higher in DM-I/R + RAPA as compared to DM-I/R group (LVEF: 45 ± 2.1%, LVFS: 25 ± 1.3%, SV: 650 ± 45 µL, CO: 179 ± 8.7 mL/min, respectively). Although there were no significant differences in LV end-diastolic diameter (LVEDD) and heart rate between groups, the LV end-systolic diameter (LVESD) was significantly reduced in the DM-I/R + RAPA group after I/R injury (9.1 ± 0.5 mm) as compared to DM-I/R group (11.35 ± 0.4 mm).

### 2.2. Rapamycin Reduces Cardiac Fibrosis after I/R Injury

Cardiac fibrosis was assessed by H&E staining and Picrosirius red staining in the post-I/R hearts. As shown in Figure 3, the disruption of the tissue architecture and fibrosis were clearly apparent in the DM-I/R group. Treatment with RAPA improved the tissue architecture and significantly reduced fibrosis following I/R injury in the DM-I/R + RAPA (19.02 ± 1.6%) as compared to the DM-I/R group (26.05 ± 1.2%) (Figure 3A,B). FITC-labeled wheat germ agglutinin (WGA, binder of glycoproteins on the cell membrane) staining was performed to label the skeletal and cardiac sarcolemma to determine cross-sectional area or myocyte density and size. The cardiomyocyte size/cross-sectional area was significantly reduced with RAPA treatment (DM-I/R + RAPA) as compared to the vehicle treated group (DM-I/R) (Figure 4).

The anti-fibrotic and anti-proliferative effects of RAPA at the cellular level was also examined in neonatal rat cardiac fibroblasts after phenylephrine (PE) treatment. Treatment with PE (100 µM for 48 h) stimulated fibroblast cell proliferation, which was inhibited by co-treatment with RAPA (100 nM) with reduction of cell size (Appendix A). Moreover, RAPA also suppressed the cell size of human ventricular AC16 cardiomyocytes after 48 h of treatment with PE (100 µM) under hyperglycemic condition (Appendix A).

### 2.3. Rapamycin Alters the Expression of Multiple Fibrosis Markers

Western blot analysis showed that treatment with RAPA inhibited the post-I/R increase in several cardiac fibrosis markers including TGF-β1, Galectin-3, MYH, p-SMAD, but induced tissue inhibitor of metalloproteinases-1 (TIMP-1, Figure 5). Real-time PCR also showed that mRNA levels of TGF-β1 and collagen type I α1 (Col-1A1) were suppressed after I/R injury in the hearts of RAPA-treated diabetic rabbits as compared to vehicle controls. However, RAPA treatment significantly upregulated the mRNA levels of TIMP-1, bone morphogenetic protein 2 (BMP2), Natriuretic Peptide A (Nppa) and cardiac α-myosin heavy chain (Myh6) following 10 days of reperfusion (Figure 6).

### 2.4. Rapamycin Attenuates NLRP3-Inflammasome following Ischemia/Reperfusion Injury

Immunofluorescence staining for apoptosis speck-like protein with a caspase recruitment domain (ASC) showed substantial aggregation (red fluorescence: Alexa 594) in the troponin positive cardiomyocytes (green fluorescence: Alexa 488) in the diabetic hearts after I/R injury (DM-I/R, peri-infarct region, Figure 7). RAPA treatment attenuated NLRP3-inflammasome formation as indicated by suppression of ASC aggregation in the cardiomyocytes (DM-I/R + RAPA). Treatment with RAPA also suppressed NLRP3 level (Figure 8) and diminished caspase 1 activity (Figure 9) in cardiomyocytes (DM-I/R + RAPA) as compared to controls (DM-I/R + RAPA).

## 3. Discussion

We have previously shown that low-dose treatment with RAPA protects against I/R injury in normal and diabetic mice and rabbits [5,6,9,10,11,16]. In the present study, we further investigated the effect of RAPA in inhibiting fibrosis, quelling inflammation, and attenuating cardiac remodeling in diabetic rabbits following reperfusion for 10 days after ischemia. The results showed that a single-dose treatment with RAPA at reperfusion attenuated LV systolic dysfunction as shown by the preservation of LVEF and LVFS and improvement of SV and CO. The LVESD is a sensitive marker of cardiac remodeling, as it reflects ventricular size and cardiac function [21]. The LVESD was significantly reduced in RAPA-treated diabetic rabbits as compared to vehicle-treated diabetic rabbits (Figure 2).

Following myocardial I/R injury, the cardiac fibroblasts transdifferentiate into highly active myofibroblasts, which replace the damaged and dead cardiomyocytes by forming scar tissue to prevent infarct expansion and ventricular dilation. Although the initial fibrotic response is an adaptive and protective mechanism, persistent fibrotic activity with exaggerated fibrosis results in irreversible adverse ventricular remodeling and progressive impairment of cardiac function, which leads to the development of heart failure over time [22,23]. Considering the high prevalence of diabetes-associated heart failure, accentuated adverse fibrotic myocardial remodeling augments the incidence of heart failure with severe cardiac hypertrophy and cardiac dysfunction in patients surviving with myocardial infarction (MI) [24]. Thus, attenuation of fibrotic remodeling following MI may be an attractive therapeutic strategy to lessen the risk of heart failure under diabetic conditions. Unfortunately, patients following MI still encounter persistent cardiac fibrosis even when receiving the standard anti-fibrotic pharmacological intervention [25]. Therefore, it is an ever-growing clinical challenge to implement potential pharmacotherapies to inhibit the proliferation/activation of cardiac fibroblasts after MI. Our results showed that cardiac fibrosis was significantly suppressed in the RAPA-treated diabetic rabbit following I/R injury (Figure 3). RAPA-treated hearts displayed reduced cardiomyocyte cross-sectional area compared with the vehicle-treated heart (Figure 4). Phenylephrine (PE)-induced proliferation of neonatal rat cardiac fibroblasts with the reduction of cellular size further supported the anti-fibrotic and anti-hypertrophic effects of RAPA (Appendix A). The anti-hypertrophic effect of RAPA also confirmed by the suppression of PE-induced increased cell size in AC-16 under hyperglycemic condition (Appendix A).

RAPA also regulates several pro-fibrotic factors including TGF-β, which is a key cytokine/growth factor inducing fibroblast activation and differentiation into myofibroblasts that secrete extracellular matrix proteins (ECM). TGF-β1, the predominant form of TGF-β, plays a critical role in the pathogenesis of cardiac fibrotic and hypertrophic remodeling [26]. TGF-β1 binds and activates the TGF-β receptor I (TGFBR1) and type II receptor (TGFBR2), which induce phosphorylation of the Smad2/3 pathway [27]. TGF-β/Smad signaling leads to the proliferation of the fibroblasts and extracellular collagen deposition following I/R injury [28,29,30]. In the present study, RAPA treatment significantly reduced the expression of TGF-β1 and Smad following I/R injury as compared to vehicle treated diabetic rabbits (Figure 5). Phosphorylation of Smad was also reduced with RAPA treatment although the ratio of p-Smad to total Smad was not altered between groups. mRNA expression of TGF-β1 was also suppressed with RAPA treatment as compared to vehicle treated diabetic rabbits (Figure 6).

We also showed that Galactin-3 expression was significantly reduced following RAPA treatment (Figure 5). Galactin-3 is a pro-fibrotic marker, which belongs to a family of soluble β-galactoside-binding lectins and is primarily expressed in myocardial fibroblasts as well as macrophages. Galectin-3 stimulates cardiac inflammation, fibroblast proliferation, collagen deposition, and ventricular dysfunction in failing hearts [31,32]. Elevated Galectin-3 level is a predictor of the development of acute ischemic events and heart failure in diabetic patients as well as diabetic cardiomyopathy with depressed cardiac function [33]. In the current study, the anti-fibrotic effect of RAPA was further supported by the suppression of Col-1A1 mRNA level and increased expression of TIMP-1 as compared to vehicle treated rabbits after I/R injury (Figure 5 and Figure 6). Cardiac fibrosis is associated with the predominance of the synthesis of collagen types I (Col-I) and collagen type III (Col-III) over their degradation, which results in the excessive accumulation of Col-I and Col-III within the myocardium [34]. The balance between the synthesis and degradation of Col-I and Col-III has a major impact on the myocardial remodeling and function in diabetic patients with heart failure [35]. Following I/R injury, hyper-activated cardiac fibroblasts produce excessive matrix metalloproteinases (MMPs) with alteration of TIMPs, which directly impact the ECM turnover and homeostasis. Accordingly, MMP/TIMP balance is associated with adverse cardiac fibrosis and subsequent heart failure after MI [36,37]. Deficiency of TIMP-1 in mice exacerbates LV remodeling following MI, which is abrogated with pharmacological MMP inhibition [38,39]. These studies indicate that increased MMP activity due to the TIMP-1 deficiency in the heart may promote adverse myocardial remodeling after MI.

We have previously shown downregulation of α-MHC in diabetic mice hearts, which was restored after chronic RAPA treatment [13]. In the present study, RAPA suppressed the post-I/R expression of β-MHC, but increased MyH6 mRNA level (Figure 5 and Figure 6), which is consistent with our previously reported results [13]. The shifting of two cardiac myosin heavy-chain isoforms (α-MHC and β-MHC) contributes to the development of dilated and hypertrophic cardiomyopathy as well as heart failure [40,41,42]. MYH6 encodes α-MHC, the key contractile protein with higher ATPase activity [43] and MYH7 encodes β-MHC. The expression of α-MHC, a critical isoform for normal myocardial function, is repressed after I/R injury and heart failure in mammalian myocardium, whereas the expression of β-MHC is stimulated [44,45]. The human non-failing myocardium expresses detectable levels of α-MHC mRNA, which decreases during the transition to heart failure. Dilated and ischemic cardiomyopathic human ventricles predominantly express β-MHC with very low levels of α-MHC [46]. Our data suggest that RAPA may restore the cardiac function and suppress cardiac fibrosis in the diabetic heart following I/R injury and remodeling by regulating the shifting of α-MHC to β-MHC.

In the present study, we also observed an increase in the mRNA level of natriuretic peptide type A (NPPA/atrial natriuretic peptide) following RAPA treatment (Figure 6). NPPA, secreted by cardiomyocytes in response to various stimuli, regulates cardiovascular homeostasis by limiting cardiomyocyte hypertrophy, fibrosis, remodeling and cardiac dysfunction [47,48,49,50,51]. Clinical studies also confirm the beneficial effects of NPPA infusion on myocardial reperfusion injury and cardiac remodeling [52,53,54]. Low ANP levels are associated with the development of diabetes and insulin resistance through the activation of the renin–angiotensin system [55,56,57].

Our results also showed that treatment with RAPA elevated the mRNA level of BMP2 (Figure 6), which may potentially contribute to the anti-fibrotic/remodeling effect of RAPA. BMPs, a highly conserved subgroup of the TGF-β superfamily, critically regulate glucose homeostasis and insulin resistance in the setting of diabetes [58]. BMPs (BMP2, BMP4, and BMP7) elicit anti-atherogenic and anti-inflammatory effects, while reducing LV remodeling and preserving cardiac function in diabetic cardiomyopathy. BMP2 and BMP6 enhanced insulin-mediated glucose uptake in both insulin-sensitive and -insensitive adipocytes [59]. Obese diabetic mice with chronic hyperglycemia and cardiac dysfunction showed elevated expression of LV TGF-ß and Smad3 protein, while Smad1/5 and BMP2 protein levels were reduced [60].

The inflammatory responses following MI stimulate myocardial collagen production and aggravate cardiac fibrosis, where the NLRP3 inflammasome is a critical determinant [61]. NLRP3 inflammasome, a multiple-protein complex, consists of NLRP3 (cryopyrin), ASC, and procaspase-1 protein [62]. In its active state, NLRP3 interacts with ASC and activates procaspase-1 by autocatalytic cleavage to the active caspase-1 that leads to maturation and secretion of IL-1β and IL-18 [63]. Inflammasome-mediated caspase-1 activation results in a proinflammatory form of programmed cell death, pyroptosis, a catastrophic form of cell demise in cardiovascular diseases [64,65]. In diabetes, hyperglycemia-induced oxidative stress due to myocardial mitochondrial perturbations triggers inflammatory signaling factors, such as nuclear factor-κB (NF-κB) and NLRP3 inflammasome, which contribute to the severity of MI-related injury, adverse cardiac remodeling, and heart failure [66,67]. Activation of the NLRP3 inflammasome promotes TGF-β signaling and induces cardiac fibrosis in diabetic hearts subjected to MI [17]. Several studies have suggested that the activation of the NLRP3 inflammasome plays a role in the development of insulin resistance in T2D and high fat diet-induced diabetes [68,69]. Genetic ablation or pharmacological inhibitors of NLRP3 or ASC and caspase-1 improve glucose tolerance and insulin sensitivity in HFD-fed mice and prevent post-MI cardiac enlargement and limit infarct size [4,68,70,71]. Interestingly, in the present study, substantial ASC aggregation was identified in cardiomyocytes in diabetic hearts following I/R injury, which was reduced with RAPA treatment (Figure 7). Treatment with RAPA suppressed NLRP3 expression (Figure 8) and active caspase-1 level (Figure 9). These results suggest that RAPA may also prevent caspase-1 activation-mediated pyroptosis, thereby averting TGF-β-induced ECM synthesis and collagen production, as well as impeding adverse remodeling in diabetic hearts with MI.

In the present study, we used our established model of type 1 diabetes with the procedure of I/R performed in the conscious rabbits [72], which has several advantages over small diabetic animal models, rats and mice. Rat and mouse hearts display marked physiological differences compared to the human heart [73], and high mortality is associated with surgical complications for coronary artery ligation under diabetic conditions [74,75]. Rabbit hearts closely resemble the human heart in terms of cardiovascular anatomy, cardiac function, cardiac metabolism, electrophysiology, coronary artery distribution, and collateral circulation after acute MI [76,77,78]. The larger size of rabbits greatly facilitates cardiac surgical and postsurgical procedures compared to rats and mice. For multiple assays, the size and dimensions of a rabbit heart are large enough to mount the transverse section of the entire heart slice in a standard glass slide for immunofluorescence studies as well as further molecular studies. As compared to highly expensive and labor-intensive larger animals, such as dogs, sheep and swine, the lower phylogenetic scale, longer life span, low cost, high reproductivity, and ease of handling of rabbit I/R models provide reliable data with higher confidence for cardiac research [78,79]. Therefore, the potential clinical significance of the current study is high.

## 4. Material and Methods

### 4.1. Induction of Diabetes in Rabbit

New Zealand male rabbits (age: 3–4 months; weight: 2.8–3.0 kg; *n* = 14) were purchased from Robinson Services Incorporated (RSI, Mocksville, NC, USA). All animal experiments were performed in accordance with USDA regulations and were approved by the Institutional Animal Care and Use Committee at the Virginia Commonwealth University. Diabetes in rabbits was induced as described in our previous publication [16,80]. Briefly, alloxan monohydrate (125 mg/kg, Sigma Aldrich, St. Louis, MO, USA) was administered via ear vein for 10 min in lightly sedated rabbits (*n* = 24) with ketamine (35 mg/kg), Xylazine (5 mg/kg) and Atropine (5 mg/kg). Blood glucose level was carefully monitored using Contour glucose meter (Bayer Corporation, Whippany, NJ, USA ) to prevent hypoglycemic shock (blood glucose level < 70 mg/dL; by supplementing 5% dextrose, i.m. and 20% glucose in drinking water for 3 days) or hyperglycemic levels (blood glucose level > 400 mg/dL; by treating with insulin, i.m., 1–2 U/kg, Novalin-R, Nova Nordisk Pharmaceutical, Plainsboro, NJ, USA). After 4 weeks of alloxan treatment, animals with blood glucose consistently above 220 mg/dL were considered diabetic and included in the protocol for further experiments.

### 4.2. Conscious Ischemia/Reperfusion Injury

Diabetic rabbits were randomized into two groups: DM-I/R (*n* = 7) and DM + I/R treated with RAPA (DM-I/R + RAPA, *n* = 7). The detailed procedure of the rabbit model of conscious I/R injury has been previously described [16,80,81]. Briefly, after anesthetizing by intramuscular (i.m.) administration of ketamine (35 mg/kg), xylazine (5 mg/kg) and atropine (5 mg/kg), the rabbit was intubated and connected to a ventilator (28–30 breaths/min). After exposing the heart through a left thoracotomy in the fourth intercostal space, a hydraulic balloon occluder was placed on top of the coronary artery and secured with the 3-0 silk on the anterior LV wall and the chest wall was closed. Seven days after successful implantation of balloon occluder, the sedated rabbits were subjected to a 45-min conscious ischemia by inflating the balloon occluder, followed by 10 days of reperfusion (by deflating the balloon occlude). To alleviate discomfort, rabbits received ketoprofen (3.0 mg/kg; i.m.) and diazepam (4 mg/kg; i.m.) 2 h before inflating the balloon for inducing ischemia. Based on the treatment groups, 5 min before the onset of reperfusion, rabbits were infused with either RAPA (0.25 mg/kg, i.v.) or DMSO (vehicle) via marginal ear vein catheter.

### 4.3. Assessment of Cardiac Function

Echocardiographic measurements were performed using a Vevo2100TM (VisualSonics Inc., Toronto, ON, Canada) following 10 days of reperfusion of the sedated (inhaled isoflurane 2.5%) diabetic rabbits treated with RAPA or vehicle. Two operators, blinded to rabbit cohort allocation, performed repeated rounds of echocardiography to minimize inter- and intra-observer variations. Short and long parasternal views were obtained to measure cavity dimensions. The percentage LVEF, LVFS, LVEDD, LVESD, SV, CO and HR were calculated by tracing the end- and epicardial boarder during contraction [82]. The obtained images were analyzed using Vevo LAB 3.2.0 software.

### 4.4. Cardiac Fibrosis Measurement

Cardiac fibrosis was evaluated by picro-sirius red staining in the risk area of LV tissue that was fixed in 4% paraformaldehyde and then embedded into paraffin blocks to prepare 5-µm-thick sections. The tissue sections were deparaffinized by incubation in Xylene and different concentrations (100%, 95%, 90% and 80%) of ethanol. The slides were later washed in distilled water for 5 min and were counter stained for nuclei by incubating with Weigert’s hematoxylin for 10 min and immediately washed in running tap water for 10 min. The collagen content of the tissue sections was analyzed by staining with picro-sirius red staining (Sigma-Aldrich, St. Louis, MO, USA, Cat: No:365548-5G) for 60 min and immediately washed two times with acidified water. The slides containing the tissues were dehydrated by incubating in 100% ethanol for three times. The slides were then mounted using resinous medium and scanned using Vectra Phenoimager HT system. Cardiac fibrosis was quantified using Image J software (NIH, Bethesda, Rockville, MD, USA).

### 4.5. Wheat Germ Agglutinin (WGA) Staining

Briefly, the tissue sections were deparaffinized by incubation in Xylene and different concentrations (100%, 95%, 90% and 80%) of ethanol. Cell size was measured by staining the LV sections with Lectin from triticum vulgaris conjugated with FITC (# L4895, Sigma-Aldrich, St. Louis, MO, USA) at a final concentration of 30 µg/mL dissolved in PBS for 30 min protected from light. The slides were then washed three times in PBS and mounted using ProLong™ Diamond Antifade Mounting with DAPI (Cat # P36966, ThermoFisher Scientific, Waltham, MA, USA). After taking pictures under a confocal microscope (Nikon D-Eclipse C1 confocal microscope), the cell size was quantified using Image J software (NIH, Bethesda, MD, USA).

### 4.6. Western Blot Analysis

Total soluble protein was extracted from the frozen LV tissues with lysis buffer (Cell Signaling, Danvers, MA, USA). Protein samples (50 µg) were resolved by SDS-PAGE, transferred onto a nitrocellulose membrane. Membranes were incubated overnight with mouse monoclonal antibody specific for TGFβ1 (sc-52893), galectin-3 (B2C10; sc-32790), TIMP-1 (2A5; sc-21734), MYH (b-5; sc-376157) and Smad 1/2/3 (H-2; sc-7960), purchased from Santa Cruz Biotechnology, Inc., Dallas, TX, USA. GAPDH (D4C6R; Mouse mAb #97166) was purchased from Cell Signaling, Danvers, MA, USA. The blots were then incubated for 1 h with anti-mouse secondary horseradish peroxidase-conjugated antibody (GE healthcare, Bensalem, PA, USA) and developed using Western Lightning Plus–ECL substrate (PerkinElmer, Waltham, MA, USA) and BioRad ChemiDoc^TM^ MP Imaging system. The densitometry analysis was performed to quantitate the intensity of the protein band using Image J software (NIH, Bethesda, MD, USA).

### 4.7. RNA Isolation and mRNA Expression

Total RNA was isolated from frozen LV tissue using RNeasy mini kit (QIAGEN Sciences, Germantown, MD, USA). The concentration and purity of isolated RNA were verified using Nanodrop ND-1000 spectrophotometer (Agilent technologies, Santa Clara, CA, USA). Total RNA (2 µg) was subjected to reverse transcription using high-capacity cDNA synthesis kit (Applied Biosystems; Waltham, MA, USA). cDNA was subjected to real-time PCR using TaqMan™ mRNA Assay kits (purchased from Applied Biosystems; Waltham, MA, USA) for quantification of TGF-β (4448892;Hs01085997_m1), Col-1A1 (4331182;Hs00164004_m1), TIMP-1 (4453320;Hs01092512_g1), BMP2 (4453320; Mm01340178_m1), Nppa (4331182; Oc03397714_g1), Myh6 (4331182; Oc03395961_m1) and normalized using GAPDH (4331182; Oc03823402_g1) using CFX96 Touch Real-Time PCR Detection System—Bio-Rad Laboratories, Hercules, CA, USA.

### 4.8. Immunofluorescence Staining

Risk area of LV tissue (*n* = 5 per group) was immediately fixed in 4% paraformaldehyde and then embedded into paraffin blocks to prepare 5µm thick sections. For immunofluorescence staining, sections were deparaffinized by incubation in Xylene and different concentrations (100%, 95%, 90% and 80%) of ethanol. The slides were later washed in distilled water for 5 min and proceeded for immunostaining. The sections were permeabilized by incubation in warm citrate buffer (pH 6.0) for 30 min and then blocked with 5% goat serum in PBS buffer. The slides were then incubated overnight with specific antibodies: NLRP3 (D4D8T) rabbit mAb (#15101), Caspase-1 (E9R2D) rabbit mAb (#83383), ASC/TMS1 (D2W8U) rabbit mAb (#67824) [Cell Signaling Technology, Danvers, MA] and Anti-Troponin T mAb (T6277) (Sigma-Aldrich, Inc., St. Louis, MO, USA). The slides were washed two times in PBS for 5 min each and incubated in secondary antibody conjugated with Alexa Fluor Anti-mouse IgG (H + L), F(ab′)2 Fragment (Alexa Fluor^®^ 488 Conjugate) (Cat #4408) or antibody conjugated to Alexa Fluor Anti-rabbit IgG (H + L), F(abimpro′)2 Fragment (Alexa Fluor^®^ 594 Conjugate) (Cat #8889) (obtained from Cell Signaling Technology, Danvers, MA) for 2 h at room temperature. The slides were later washed twice in PBS and mounted using ProLong™ Diamond Antifade Mounting with DAPI (Cat # P36966, ThermoFisher Scientific, Waltham, MA, USA). After taking pictures (five random images per cardiac section) under a confocal microscope (Nikon D-Eclipse C1 confocal microscope), the ASC speck and DAPI-positive nuclei were quantified using Image J software (NIH, Bethesda, MD, USA) and data are presented as the percent of ASC speck with respect to DAPI-positive nuclei. NLRP3 and active caspase-1 fluorescence were quantified manually with respect to DAPI-positive nuclei and data are presented as fold change.

## 5. Conclusions

We provided evidence that treatment of diabetic rabbits with a single dose of RAPA at the time of reperfusion following lethal ischemia is highly effective in restoring cardiac function through attenuation of fibrosis and inflammation. Specifically, RAPA suppressed several pro-fibrotic markers and prevented NLRP3 inflammasome formation in diabetic hearts with MI. While the long-term beneficial effects of RAPA beyond 10 days of reperfusion were not investigated, the current results suggest that the mTOR pathway could be a potential therapeutic target for treatment of adverse post-infarct myocardial remodeling and inflammation in diabetic patients.

## Figures and Tables

**Figure 1 ijms-24-08998-f001:**
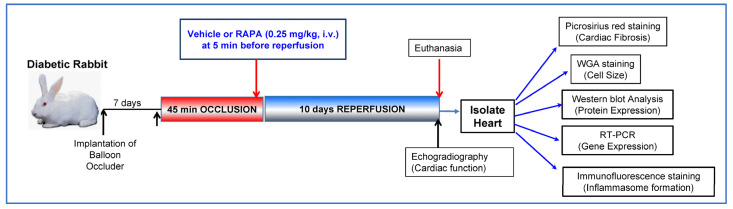
Experimental protocol for myocardial ischemia and reperfusion injury in conscious diabetic rabbit showing time points for experimental procedures, administration of drugs, measurement of cardiac function and collection of heart tissue for determination of various molecular markers.

**Figure 2 ijms-24-08998-f002:**
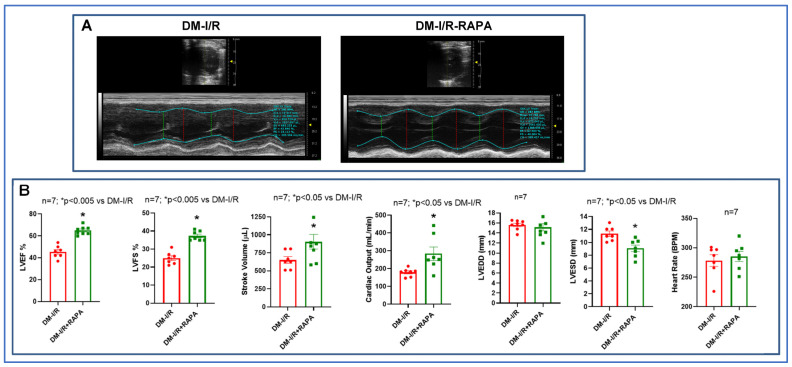
Post-ischemic cardiac function in diabetic rabbits. (**A**) Representative cardiac images of parasternal view (M-mode ultrasound) with the preview of B-mode image (inner panel) of diabetic rabbit following 45 min ischemia and 10 days of reperfusion (DM-I/R) with and without treatment with rapamycin (RAPA) at the onset of reperfusion (DM-I/R + RAPA). (**B**) Percentage of left ventricular ejection fraction (LVEF) and fractional shortening (LVFS), stroke volume (SV), cardiac output (CO), LV end diastolic diameter (LVEDD), LV end systolic diameter (LVESD), and heart rate of DM-I/R and DM-I/R + RAPA rabbits.

**Figure 3 ijms-24-08998-f003:**
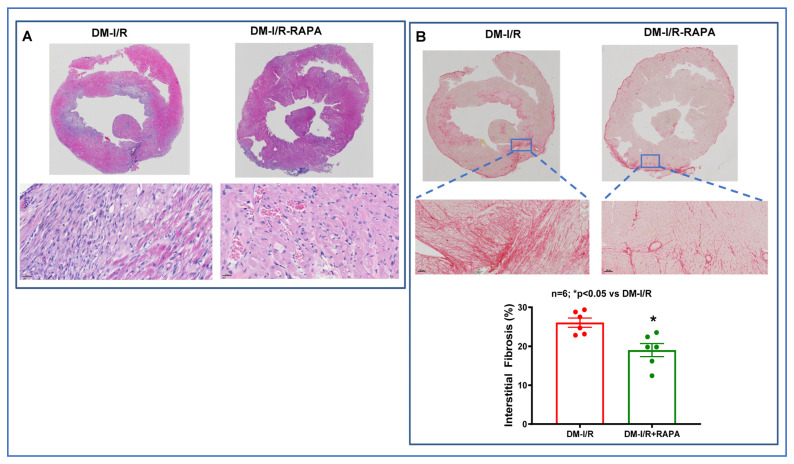
Myocardial histology and fibrosis in diabetic rabbits following 45 min ischemia and 10 days of reperfusion (DM-I/R). (**A**) Hematoxylin and eosin staining and (**B**) Picrosirius red staining of sections of the hearts from DM-I/R and rapamycin (RAPA)-treated (DM-I/R + RAPA) rabbits. Bottom: Percentage of Picrosirius red-positive cardiac cross-sectional area.

**Figure 4 ijms-24-08998-f004:**
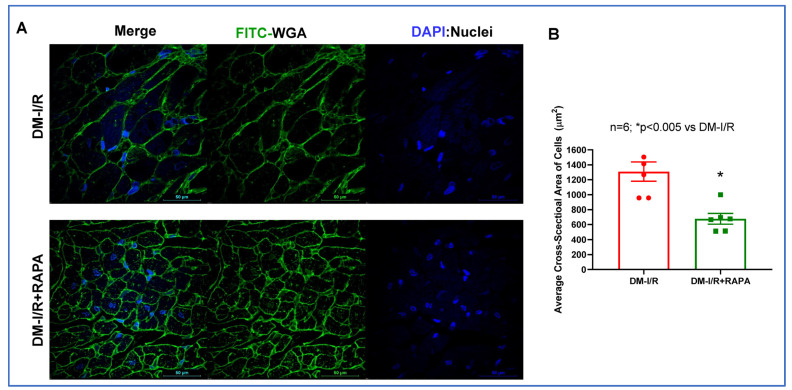
(**A**) Representative pictures of the FITC-labeled WGA (FITC-WGA) staining in diabetic rabbits following 45 min of ischemia and 10 days of reperfusion (DM-I/R). Rapamycin (RAPA) was administered at 5 min before the onset of reperfusion (DM-I/R + RAPA). Cell membrane is in green color (FITC-WGA) and nuclei are in blue color (DAPI staining). (**B**) Quantification of average cardiomyocyte cross-sectional area.

**Figure 5 ijms-24-08998-f005:**
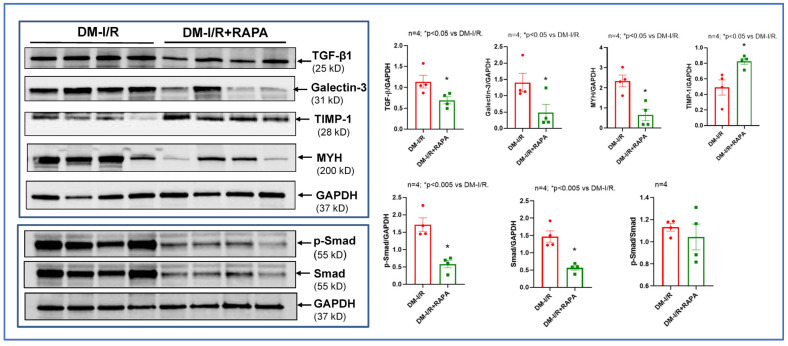
Effect of Rapamycin (RAPA) treatment on the expression of cardiac fibrosis markers in diabetic rabbits. Representative immunoblots (**Left** panel) and densitometry analysis of TGF-β, Galectin-3, myosin heavy chain (β-MHC), tissue inhibitor of metalloproteinases (TIMP-1) and p-Smad and total Smad expression (**Right** panel) in myocardium of diabetic rabbits following 45 min of ischemia and 10 days of reperfusion (DM-I/R) with/without RAPA treatment at the onset of reperfusion (DM-I/R + RAPA).

**Figure 6 ijms-24-08998-f006:**
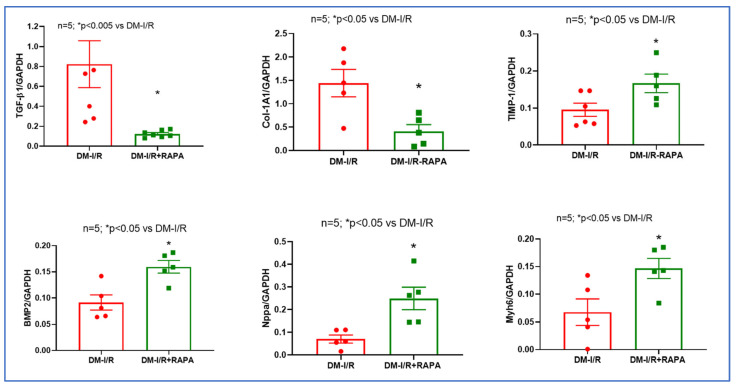
Effect of Rapamycin (RAPA) treatment on cardiac fibrosis markers diabetic rabbits. The mRNA levels of TGF-β, Galectin-3, myosin heavy chain (β-MHC), tissue inhibitor of metalloproteinases (TIMP-1) and p-Smad and total Smad expression (**Right** panel) in the hearts of diabetic rabbits following 45 min of ischemia and 10 days of reperfusion (DM-I/R) with/without treatment with RAPA at the onset of reperfusion (DM-I/R + RAPA).

**Figure 7 ijms-24-08998-f007:**
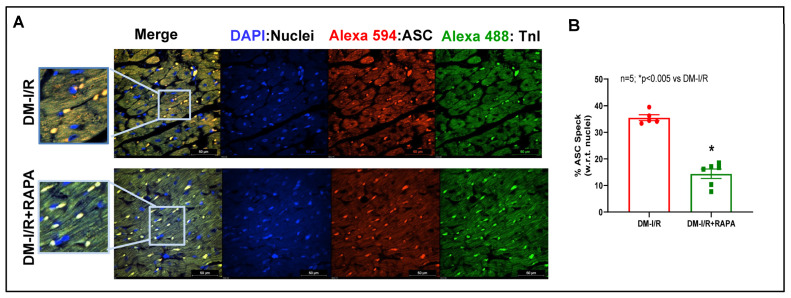
(**A**) Representative pictures of ASC aggregation (red: Alexa 594) in diabetic rabbit hearts after 45 min ischemia and 10 days of reperfusion (DM-I/R). Rapamycin (RAPA) was administered at 5 min before the onset of reperfusion (DM-I/R + RAPA). Cardiomyocytes are in green color (Troponin I: Alexa 488) and nuclei are in blue color (DAPI staining). (**B**) Percentage of cells with ASC aggregation with respect to total nuclei.

**Figure 8 ijms-24-08998-f008:**
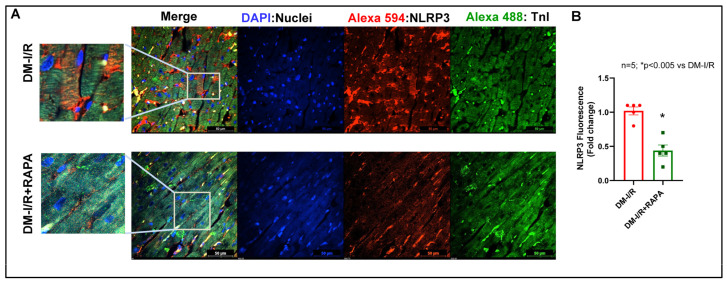
(**A**) Representative pictures of NLRP3 expression (red: Alexa 594) in diabetic rabbit hearts after 45 min ischemia and 10 days of reperfusion (DM-I/R). Rapamycin (RAPA) was administered at 5 min before the onset of reperfusion (DM-I/R + RAPA). Cardiomyocytes are in green color (Troponin I: Alexa 488) and nuclei are in blue color (DAPI staining). (**B**) NLRP3 positive fluorescence was quantified with respect to total nuclei and presented as fold change.

**Figure 9 ijms-24-08998-f009:**
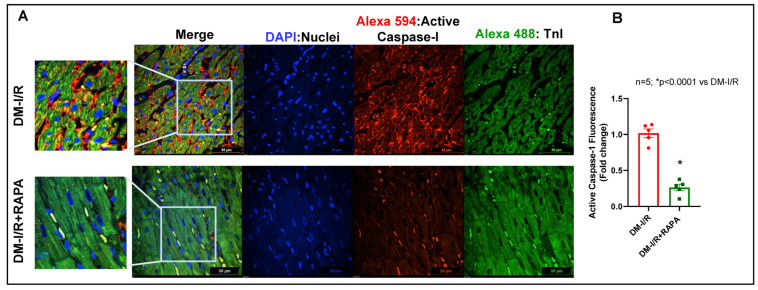
(**A**) Representative pictures of active caspase-I (red fluorescence) in diabetic rabbit hearts after 45 min ischemia and 10 days of reperfusion (DM-I/R). Rapamycin was administered at 5 min before the onset of reperfusion (DM-I/R + RAPA). Cardiomyocytes are in green color (Troponin I: Alexa 488) and nuclei are in blue color (DAPI staining). (**B**) Active caspase-1 positive fluorescence was quantified with respect to total nuclei and presented as fold change.

## Data Availability

The data presented in this study are available upon request from the corresponding author.

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
