# Peer review of "Single-Dose Treatment with Rapamycin Preserves Post-Ischemic Cardiac Function through Attenuation of Fibrosis and Inflammation in Diabetic Rabbit"

_ijms, 2023, doi:10.3390/ijms24108998_

Round 1
Reviewer 1 Report
This is a very well done work taking the groups work on rapamycin to the next step in preclinical models. The same group has previously developed this rabbit diabetes and conscious infarct model, and has published that rapamycin treatment reduces infarct size in the model. This work establishes a definitive clinically relevant result and probes the molecular mechanisms involved in the protection against MI. This group has previously developed this rabbit diabetic model and shown that rapamycin reduced infarct size 6 hours after reperfusion when given during occlusion, a clinically relevant timing. Herein, they show that there is reduced infarct size and improves cardiac function ten days post-MI, after giving the animals a single injection to rapamycin at repercussion. This exciting result extends the clinical relevance to the work to date. Mechanistic studies show that mTor signaling is increased in the hearts of diabetic rabbits, and that rapamycin treatment inhibits fibrotic markers, attenuated NLRP3 inflammasome signaling and preserves expression of the alpha-MyHC while reducing expression of the beta-MyHC, hallmarks of adverse remodeling and reduced function. The results support that repercussion therapy with rapamycin may have clinical relevance through these mechanisms, and extends the relevance to more than the acute effect of MI. Overall, the work is rigorous and well-presented. The data supports the interpretations. The one issue I see is that the micrographic data in Figs 7, 8, 9 is representative, but we are not told in the figure legend or methods how many animals were used in thee studies, how many slides interpreted and how visualized sections were selected. These details need to be added.
Author Response
- The one issue I see is that the micrographic data in Figs 7, 8, 9 is representative, but we are not told in the figure legend or methods how many animals were used in these studies, how many slides interpreted and how visualized sections were selected. These details need to be added.
Response:
Thank you for your insightful suggestions to improve the in-depth findings of the present study. For all immunofluorescence studies, we used five cardiac sections (risk area of LV tissue) per group (DM-I/R and DM-I/R+RAPA). As per the suggestion, we have now specified all the details in the method section (under section 4.8. Immunofluorescence staining) as well as in Figure Legends (Figure 7, 8, 9) as state below. We also semi-quantified the percentage of ASC speck with respect to DAPI-positive nuclei and the fold changes of NLRP3 and active caspase-1 fluorescence and presented in the respective figures.
“After taking pictures (randomly five images per cardiac section) under a confocal micro-scope (Nikon D-Eclipse C1 confocal microscope), the ASC speck and DAPI-positive nuclei were quantified using Image J software (NIH, Bethesda, MD) and data was presented as the percent of ASC speck with respect to DAPI-positive nuclei. NLRP3 and active caspa-se-1 fluorescence were quantified manually with respect to DAPI-positive nuclei and data were presented as fold change.”
We greatly appreciate the excellent comments for improvement of the manuscript.

Reviewer 2 Report
Using a diabetic rabbit model, authors show that that mTOR inhibition with a single dose of RAPA has long-lasting effect in protecting the diabetic myocardium against I/R-associated fibrotic and inflammatory mediators of injury. The studies are well done and the data discussed within the confines of experimental design. These findings are of high human relevance. There are minor concerns in this study.
1. Were all the animal receiving insulin to manage excess hyperglycemia in this study?
2. In Figures 2 and 3, does "the onset of reperfusion" in the figure legend denote 10 days of reperfusion. Please clarify, other figures indicate 10 days of reperfusion.
3. In Figures 2B, 5, and 6, please indicate all the data points in the bar graph.
Author Response
We are very pleased with the excellent reviews of our manuscript entitled " Single-Dose Treatment with Rapamycin Preserves Post-Ischemic Cardiac Function through Attenuation of Fibrosis and Inflammation in Diabetic Rabbit," (ijms-2360748). Based on the comments from all the reviewers, we carefully revised the paper. All the new text for revision has been indicated in blue. Following is the itemized response to each of their comments:
Reviewer #2
- Were all the animal receiving insulin to manage excess hyperglycemia in this study?
Response: Thanks for this important clarification. Only those rabbits with blood glucose level above 400 mg/dL were treated with insulin. Detailed protocol has been published in STAR Protocol (Samidurai, A., Ockaili, R., Cain, C., Roh, S. K., Filippone, S. M., Kraskauskas, D., Kukreja, R. C. & Das, A. (2021) Preclinical model of type 1 diabetes and myocardial ischemia/reperfusion injury in conscious rabbits-demonstration of cardioprotection with rapamycin, STAR Protoc. 2, 100772).
- In Figures 2 and 3, does "the onset of reperfusion" in the figure legend denote 10 days of reperfusion. Please clarify, other figures indicate 10 days of reperfusion.
Response: Very important point which we left out due to the oversight. As per the suggestion, we have clarified in the legends of Figures 2 and 3 which reads as follows: “After 45 min of ischemia and 10 days of reperfusion, cardiac function was measured and hearts (Figure 2) were collected for histology and fibrosis measurements (Figure 3). All animals were subjected to 45 min ischemia and 10 days of reperfusion”.
Also, we have modified the legends of other figures to clarify 45 min ischemia and 10 days reperfusion. Many thanks!
- In Figures 2B, 5, and 6, please indicate all the data points in the bar graph.
Response: We indicated all the data points in the bar graphs.
We greatly appreciate the excellent comments for improvement of the manuscript.
